# Optimistic Optimization of Gaussian Process Samples

**Julia Grosse**                                          *julia.grosse@uni-tuebingen.de*
*Tübingen AI Center, University of Tübingen, Germany*

**Cheng Zhang**                                          *cheng.zhang@microsoft.com*
*Microsoft Research, Cambridge, United Kingdom*

**Philipp Hennig**                                      *philipp.hennig@uni-tuebingen.de*
*Tübingen AI Center, University of Tübingen, Germany*

**Reviewed on OpenReview:** *https://openreview.net/forum?id=KQ5jI19kF3*

## Abstract

*Bayesian optimization* is a popular formalism for global optimization, but its computational costs limit it to expensive-to-evaluate functions. A competing, *computationally* more efficient, global optimization framework is *optimistic optimization*, which exploits prior knowledge about the geometry of the search space in form of a dissimilarity function. We investigate to which degree the conceptual advantages of Bayesian Optimization can be combined with the computational efficiency of optimistic optimization. By mapping the kernel to a dissimilarity, we obtain an optimistic optimization algorithm for the Bayesian Optimization setting with a run-time of up to $\mathcal{O}(N \log N)$ in terms of number of function evaluations. As a high-level take-away we find that, when using stationary kernels on objectives of low evaluation cost, optimistic optimization can be preferable over Bayesian optimization, while for strongly coupled and parametric models, Bayesian optimization can perform much better, even at low evaluation cost. As a conceptual takeaway, our results demonstrate that balancing exploration and exploitation under Gaussian process assumptions does not require computing a posterior.

## 1 Introduction

Bayesian optimization (BO) (Shahriari et al., 2015) is a popular and successful framework for global optimization. The foundation of most BO frameworks is a Gaussian Process (GP) regression method, whose kernel encodes prior knowledge about the objective function. This GP regressor provides a posterior over the objective, a probabilistic surrogate on which one can then reason about the extremum, and its location. This can be done in a variety of ways, e.g. by using the posterior to construct upper bounds (in GP-UCB Srinivas et al. (2009)), to estimate expected improvement (Močkus, 1975) or information gain (Hennig & Schuler, 2012; Wang & Jegelka, 2017).

The computational cost of BO itself is significant, and arises from at least two sources: First, exact GP inference has cost $\mathcal{O}(N^3)$, where $N$ is the number of observed function values ("samples"). Second, finding the next evaluation location requires a *continuous, numerical* optimization of the acquisition function. Since this utility function is generally multimodal, optimization should, at least in principle, be carried out on an increasingly fine grid, which contributes up to $\mathcal{O}(N^{2d})$ costs (Salgia et al., 2021). A third source of cost can be the construction of the acquisition function itself, but there are some popular choices, like GP-UCB, for which this step is essentially a trivial sum of the mean and marginal standard-deviation constructed during inference, of negligible overhead. If the cost of individual function evaluations is very high, then the overhead of BO is irrelevant. But there are scenarios in which the cost of BO *is* a concern, namely when individual function evaluations are of intermediate cost, and (perhaps as a direct consequence), the total number $N$ of evaluations is sufficiently large to make the cubic cost of GP inference noticeable. An example are settings

involving computer simulations runs, e.g. in robotics, biology, chemistry and human-computer interaction design.

Considering this "middle ground" between sample and computational efficiency, we study a competing framework for global optimization, *optimistic optimization* (OO), which has drastically lower computational overhead. OO does not require computing an explicit global posterior on the objective. However, OO can nevertheless leverage at least certain kinds of prior knowledge, captured in the form of a Lipschitz condition with respect to a pseudo-metric, or more generally a dissimilarity function. Functions from a GP prior fulfill a similar condition with respect to the canonical pseudo-metric of the GP. This in effect produces a map from a GP prior one might otherwise use in BO to a (much more time-efficient) OO model, so that prior knowledge encoded in the GP can be leveraged without the intermediate step of (cubically expensive) GP regression. For noiseless observations, the resulting OO algorithm achieves $\mathcal{O}(N \log N)$ computational cost.

In summary, we provide a practical map between BO to OO for global optimization (Section 3), resulting in a hybrid method we call GP-OO. We then contribute a formal analysis of this method in terms of regret (Section 5), also pointing out limitations in the process. Experiments (Section 6) corroborate that the proposed GP-OO can be significantly more time-efficient than classical BO methods like GP-UCB and EI in settings with low function evaluation costs.

## 2 Background

### 2.1 Bayesian Optimization (BO)

We consider maximizing a function $f : \mathcal{X} \to \mathbb{R}$. Throughout, $f^*$ denotes the maximum of the function, and $x^*$, the point where it is attained. $k : \mathcal{X} \times \mathcal{X} \to \mathbb{R}$ is a kernel function. The function $f$ is assumed to be a sample from a GP $\mathcal{GP}(\mu, k)$. We assume that the GP is centered, i.e. $\mu = 0$. One has access to observations $y_i$, where $y_i \sim f(x_i) + \mathcal{N}(0, \lambda)$. The noiseless setting amounts to the assumption $\lambda \to 0$. After each new observation the posterior over the function is updated:

$$\mu_n(x) = \mathbf{k}_n(x)^T (\mathbf{K}_n + \lambda I)^{-1} \mathbf{y}_n \tag{1}$$

$$k_n(x, x) = k(x, x) - \mathbf{k}_n(x)^T (\mathbf{K}_n + \lambda I)^{-1} \mathbf{k}_n(x), \tag{2}$$

where $\mathbf{y}_n = [y_1, ..., y_n]^T$, $\mathbf{k}_n(x) = [k(x_1, ), ..., k(x_n, )]^T$, and $\mathbf{K}_n$ is the Gram matrix with $\mathbf{K}_{nij} = k_n(x_i, x_j)$. The posterior is used to define an acquisition function $a_n(x)$ at whose maximum the function is evaluated next. For an overview of different options of acquisition functions see Frazier (2018). We will focus our analysis on the choice of GP-UCB Srinivas et al. (2009), where

$$a_n(x) = \mu_n(x) + \beta_n^{1/2} k_n(x, x)$$

and $\beta_n^{1/2}$ an appropriate constant. An advantage of GP-UCB over other choices is its comparatively simple structure, which facilitates not just implementation but also analysis. While other acquisition functions may well be more sample-efficient in concrete settings, this will not be important for what is to follow, because we are primarily concerned with run-time complexity, where GP-UCB is one of the fastest possible choices of the classical BO methods. Furthermore, it is most similar to OO from a conceptual point of view as OO also relies on upper bounds as outlined in the next paragraph.

### 2.2 Optimistic Optimization (OO)

The optimistic optimization principle is used to optimize functions that are known to fulfill a local Lipschitz assumption with respect to a dissimilarity $d : \mathcal{X} \times \mathcal{X} \to \mathbb{R}$:

$$\forall x \in \mathcal{X} : |f(x^*) - f(x)| \le d(x^*, x). \tag{3}$$

The method revolves around a hierarchical partitioning of the search space $\mathcal{X}$ that can be described by an infinite binary tree. Search can thus be implemented as a fast descent through the tree, thus only involves evaluations on a countable set of mesh points, in contrast to the local numerical optimization done within

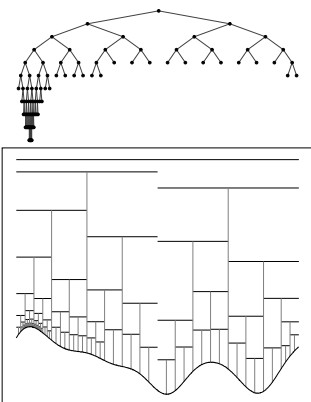

Figure 1: Optimistic optimization applied to a sample from a GP. The upper bounds are shown as horizontal bars. The vertical lines point to the evaluated locations.

Bayesian optimization. This is one of the two reasons for the drastically shorter run-time costs of OO over BO (the other being that it does not require computing a GP posterior).

The root node corresponds to the entire search space $\mathcal{X}_{(0,1)} = \mathcal{X}$ and is named $(0,1)$. Consider a node $(t,i)$ at depth $t$. The left child $(t+1, 2i-1)$ and right child $(t+1, 2i)$ represent two subregions $\mathcal{X}_{(t+1,2i-1)}$ and $\mathcal{X}_{(t+1,2i)}$ such that $\mathcal{X}_{(t,i)} = \mathcal{X}_{(t+1,2i-1)} \cup \mathcal{X}_{(t+1,2i)}$, i.e. the tree covers the entire space. To indicate that a cell was explored at iteration $n$, we refer to it as $\mathcal{X}_n$. Intuitively, it makes sense to select the cells in such a way that all points in a cell are similar to each other and all similar points are in the same cell. Formally, this can be expressed as

(a) $\forall x, y \in \mathcal{X}_{(t,i)} : \{d(x,y) < \delta(t)\}$

(b) $\exists x_{(t,i)} \in \mathcal{X}_{(t,i)} : \{y \in \mathcal{X} : d(x_{(t,i)}, y) < c\,\delta(t)\} \subset \mathcal{X}_{(t,i)}$,

where $\delta(t)$ is a decreasing sequence of diameters and $c$ is a global constant. During search, the tree is build incrementally by adding the two children of a selected node. When a new node $(t,i)$ is added, an observation is made at the center $x_{(t,i)}$ of the region $\mathcal{X}_{(t,i)}$. In each round, the leaf with the highest upper bound $U_{(t,i)} = f(x_{(t,i)}) + \delta(t)$ is selected for expansion. Since $f$ is assumed to be locally Lipschitz with respect to $d$, selecting nodes by $U_{(t,i)}$ is a valid upper bound strategy (assuming noiseless observations). The sequence of $\delta(t)$ controls exploration. Therefore, the partitioning should be chosen in such a way, that the $\delta(t)$ can be as small as possible. The optimistic optimization principle is summarized in Algorithm 1 and Figure 1. Using a binary heap, the priority queue for the leaf nodes can be realized in $\mathcal{O}(N \log N)$ if the budget $N$ is known in advance.

---

**Algorithm 1** The optimistic optimization principle.

---

1: **procedure** OPTIMISTIC OPTIMIZATION($f, \delta$)
2:     priority-queue $\leftarrow [(0,1)]$ //root node
3:     **for** $j = 1$ to $n$ **do**
4:         Select node $(t,i)$ with maximum $U_{(t,i)} = f(x_{(t,i)}) + \delta(t)$ from the priority-queue
5:         Observe $f(x_{(t+1,2i-1)})$ and $f(x_{(t+1,2i)})$ for the two children of $(t,i)$.
6:         Calculate child utilities $U_{(t+1,2i-1)} = f(x_{(t+1,2i-1)}) + \delta(t+1)$ and $U_{(t+1,2i)} = f(x_{(t+1,2i)}) + \delta(t+1)$.
7:         Add children $(t+1, 2i-1)$ and $(t+1, 2i)$ to the priority-queue based on their utilities.
8:     **end for**
9: **end procedure**

---

This algorithm is a batch method in the sense that all children of a newly explored node are evaluated in each step. We assume batch size two, but other choices are possible too and might be preferable in some

applications, e.g. when parallel evaluation is possible. A conceptual difference to the upper bound in GP-UCB is that one here works with upper bounds for entire *regions* of the search space, not for individual points in it. Also, the exploration term $\delta(t)$ in the upper bound is not updated based on new observations, but derived a priori. Nevertheless, the exploration term, aka. the uncertainty decreases during the search as the tree grows.

The hierarchical optimistic optimization principle has its origins in the Bandit setting. Bubeck et al. (2011) apply it in the noisy setting, Munos (2011) apply it in the noiseless setting and Kleinberg et al. (2008) uses it with a slightly different Lipschitz assumption. The assumptions on the dissimilarity $d$ vary, e.g. some theoretical analyses require that it additionally is a semi-metric, or metric. Munos (2011) introduces a variant of the principle, simultaneous optimistic optimization (SOO), for situations in which the dissimilarity function is unknown. Valko et al. (2013) extend this idea to the noisy setting. The popular DiRect algorithm (Jones & Martins, 2021) also belongs to the family partitioning based global optimization algorithms, but does not allow for encoding of prior knowledge on the smoothness of the function.

**An interim summary:** OO descends along a search tree, i.e. in a *discrete* sequence of steps, without numerical optimization, and only using *local* summary statistics, rather than updating a global posterior. This makes OO very fast, at least compared to Bayesian optimization. But the approach also has a downside: At least at first sight, it is not clear how to encode salient prior information about the global optimization problem into the algorithm. By contrast, many Bayesian optimization experts see the rich language of GP prior models as a key strength of the BO framework. The following section thus investigates formal connections between OO and BO. The goal is to understand to which degree the structural language of a GP prior can be translated into the algorithmic efficiency of OO.

## 3 Gaussian Process Optimistic Optimization

The policy of OO is based on measuring (dis)similarity in the input domain in terms of a pseudo-metric. It turns out that Gaussian process models – or, more precisely, kernels – can be used immediately to define such a pseudo-metric (Section 3.1). We further show how the pseudo-metric can be used to obtain upper bounds on the supremum of the cell (Section 3.2), that allow for the application of the OO principle in the BO setting. Section 3.3 is concerned with how to choose the cells and in Section 3.4 we illustrate the derived concept (GP-OO) on concrete examples.

### 3.1 Canonical pseudo-metric

The canonical pseudo-metric $d : \mathcal{X} \times \mathcal{X} \to \mathbb{R}$ for a centered GP is defined as

$$
\begin{aligned}
d(x, y) &= \mathbb{E}_{f \sim \mathcal{GP}(0,k)}[(f(x) - f(y))^2] \\
&= \sqrt{k(x, x) + k(y, y) - 2k(x, y)}.
\end{aligned}
$$

The relevance of the canonical pseudo-metric for optimization arises from the following deviation inequality (Pisier (1999), Theorem 4.7):

$$
\forall u > 0, \mathbb{P}(|f(x) - f(y)| \geq u) \leq 2 \exp\left(-\frac{u^2}{2d(x, y)^2}\right)
$$

The intriguing aspect of this inequality is that it relates distances $|f(x) - f(y)|$ "along the ordinate" with distances $d(x, y)$ "along the abscissa". This suggests that the inequality is informative for balancing exploitation ("progress along the ordinate") with exploration ("progress along the abscissa"). From a conceptual point of view, the main motivation of this work consists in deriving an adaptive optimization algorithm, i.e. an optimization algorithm that is able to trade-off exploration with exploitation based on observed functions values, relying on the above inequality instead of GP regression. The OO principle is well suited for this attempt due to its upper-bound based acquisition strategy with a simple additive structure of function observations and exploration terms.

### 3.2 Upper bound on the supremum of a cell

The first challenge in applying the optimistic optimization principle on samples of a GP consists in the probabilistic nature of the deviation inequality. To obtain a valid upper bound for the maximal deviation in a cell $\sup_{x \in \mathcal{X}_n} |f(x) - f(x_n)|$, the deviation for *all* points in the cell has to be bounded. We approximate such an upper bound by introducing two simplifications: We discretize the search space $\mathcal{X}$ into a finite number of points $\hat{\mathcal{X}}$ and we introduce an independence assumption between the $|f(x) - f(x_n)|$. Then we take a union bound approach:

$$\mathbb{P}(\sup_{x \in \hat{\mathcal{X}}_n} |f(x) - f(x_n)| \geq u) \tag{4}$$

$$\leq \sum_{x \in \hat{\mathcal{X}}_n} \mathbb{P}(|f(x) - f(x_n)| \geq u) \tag{5}$$

$$\leq 2 \sum_{x \in \hat{\mathcal{X}}_n} \exp\left(-\frac{u^2}{2d(x,y)^2}\right) \tag{6}$$

$$\leq 2|\hat{\mathcal{X}}_n| \exp(-\frac{u^2}{2\Delta(\mathcal{X}_n)^2}) \tag{7}$$

$$\text{where } \Delta(\mathcal{X}_n) = \max_{x \in \mathcal{X}_n} d(x_n, x). \tag{8}$$

The bounds have to hold at each step $n$, so we additionally take a union bound over the number of steps. This implies the following statement, that holds with high probability :

$$\forall n : \sup_{x \in \hat{\mathcal{X}}_n} |f(x) - f(x_n)| \leq \beta_n^{1/2} \Delta(\mathcal{X}_n) \tag{9}$$

where $\beta_n$ are appropriate constants specified in the Supplements. The union bound approximation will be good if the $|f(x) - f(x_n)|$ are (nearly) uncorrelated, or the size of the discretization $|\hat{\mathcal{X}}|$ is chosen sufficiently small. Otherwise the bounds are loose, which leads to over-exploration. Though approximate, this idea of neglecting correlations to simplify the calculation of the expected supremum of dependent Gaussian variables is, for example, also done in Maximum Value Entropy Search (Wang & Jegelka, 2017) for BO and in related settings (Grosse et al., 2021). The other extreme, a greedy approach with $\beta_n = 1$ has also been taken in recent work (Rando et al., 2022). For stationary kernels, we experimented with heuristical choices based on the lengthscale, see Appendix A.2.2. With generic chaining (Talagrand, 1996) it is possible to improve over the union bound approach. However, to the best of our knowledge state-of-the-art algorithms (Borst et al., 2020) to optimize for tighter bounds require polynomial run-time in the number of points per cell for arbitrary kernel functions. For special cases, like a Wiener kernel (Talagrand, 2021) or Matérn kernel functions (Shekhar & Javidi, 2018) analytical attempts to derive chaining based upper bounds exist.

### 3.3 Choosing how to partition

The second main step in applying the OO principle is to choose the cells and location of the centers in such a way that the diameters of the cells are as small as possible. For $k$ children nodes, this is a metric $k$-center problem – one of the classical NP-hard problems (Gonzalez, 1985). A greedy approximation consists in iteratively picking the $k$ centers with the largest distance to the previously picked centers, and requires $\mathcal{O}(|\hat{\mathcal{X}}|k)$ time. The greedy procedure is guaranteed to result in a 2-approximation, and there is no polynomial time algorithm doing better (unless P=NP). Working with a greedy instead of the optimal partition scheme thereby leads to an additional factor of 2 in the below regret bound, but is not harmful in the sense that the search gets stuck in a local optimum. NP-hardness also appears in the context of BO, e.g. the exploration term used in GP-UCB. And the acquisition functions of information-theoretic BO methods (Hennig & Schuler, 2012), (Wang & Jegelka, 2017) are related to the maximization of information gain, which is also a NP-hard problem.
From an implementation perspective, it is desirable to constrain the partition to axis-parallel boxes. For some kernel functions, e.g. the polynomial kernel, requirement (b) of the OO principle cannot be fulfilled

with axis parallel boxes. One can nevertheless run the algorithm, but it will clearly be less information-efficient. One may even consider a randomized choice of centers, e.g. as done in Monte Carlo Tree Search (Chaslot et al., 2008). However, it still remains to calculate the maximal distance from a point in the cell $\mathcal{X}$ to the center $x_n$. The computational complexity of this part is comparable to the numerical optimization of the acquisition function in BO. An advantage is that the domain over which one optimizes shrinks in each step. A disadvantage, though, is that if the numerical optimization is suboptimal and the maximal distance within a cell is underestimated, cells containing the optimum might get irreversibly pruned.

An important observation is that the partition scheme itself does not depend on the objective $f$, but only the kernel/distance function (how the search tree grows, however, *does* depend on $f$). This opens up the possibility of finding a good partition and the corresponding maximal distances analytically. An important class of kernel-induced metrics is formed by monotonic transformations of the Euclidean metric, i.e. $d(x, y) = g(\|x - y\|_2)$ where $g : \mathbb{R} \mapsto \mathbb{R}$ is monotonically increasing. Many kernels used in practice are in this class, e.g. the square-exponential kernel, the Matérn class of kernels, the rational-quadratic kernels and the Wiener kernel as well as sums and products thereof. We refer to this class of kernels as $\mathcal{K}$. For distances derived from kernels in $\mathcal{K}$, one can apply the following *regular* partition scheme: At each step, cut along the longest dimension in order to obtain the two children cells with respect to the euclidean metric. Use the euclidean centers as centers. A point that maximizes the distance to the center will always be one of the $2^m$ corner points in $m$ dimensions. However, one only has to calculate the distance from the center to one of them. Thus, for this type of kernels, the costs reduce to $\mathcal{O}(m)$ for the partition, or $\mathcal{O}(N \log N + N \cdot m) = \mathcal{O}(N \log N)$ in total. Kernels not in $\mathcal{K}$ are e.g. polynomial or periodic kernels.

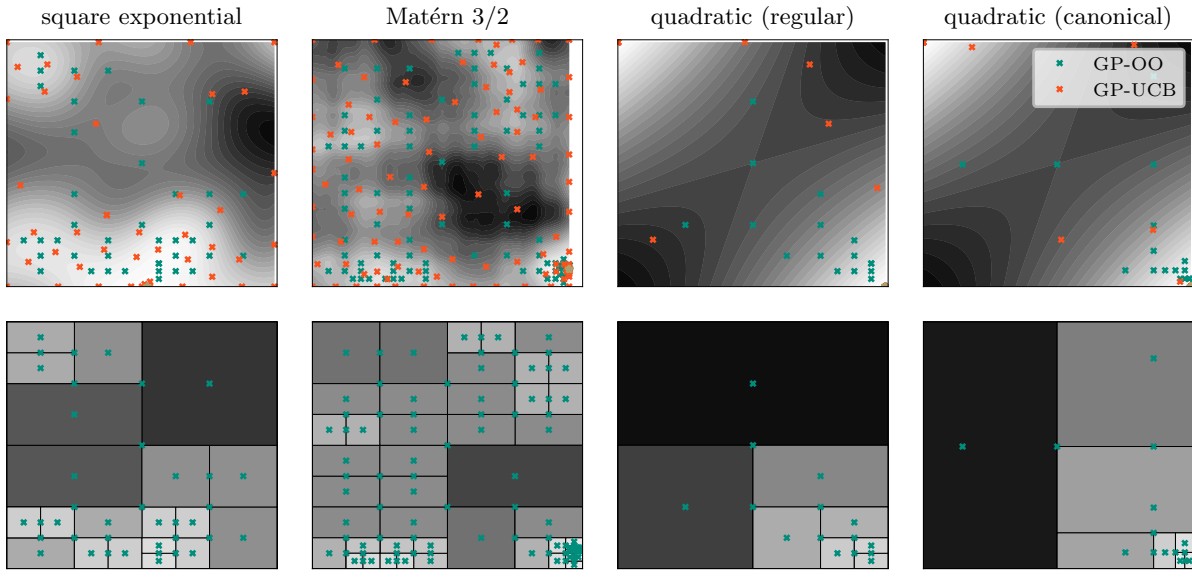

Figure 2: Lower row: Cells and diameters $\Delta$. Brighter colors indicate smaller diameters. Upper row: Sample from a GP and evaluation locations of GP-OO and GP-UCB. Bright colors indicate higher function values.

### 3.4 GP-OO

Motivated by the analysis above, we propose a variant of OO, which we call GP-OO. It consists in running Algorithm 1 with the utility $\mathcal{U}_{(n)} = f(x_{(n)}) + \beta_n^{1/2} \Delta(\mathcal{X}_n)$ in line 4, where $\Delta$ is as defined in Eq. equation 8. Figure 2 shows the algorithm running on GP samples from a square-exponential and a Matérn kernel with $\nu = 2/3$ on the domain $[0, 5]^2$, as well as from a quadratic kernel on the domain $[-1, 1]^2$. In regions with higher function values the partitions are finer. The Matérn kernel yields higher distances than the square exponential, leading to more exploration, reflecting that the samples are less smooth. By optimizing the partitions and centers of the cells with respect to the canonical pseudo-metric, larger parts of the search space can be covered while keeping the cell diameters constant as shown by the two examples with the

quadratic kernel. However, as GP-OO is restricted to evaluations on a grid, it usually requires more steps than GP-UCB to reach the optimum even if the grid is refined in the right regions.

## 4 Related Work

### 4.1 Work at the intersection of BO and OO

With the exception of Grill et al. (2018), the fundamental difference to all of this work is that we do not keep track of a GP-posterior, thus saving significant computational cost.

**Work without the canonical pseudo-metric.** BO methods have been combined with SOO (Munos (2011)), the version of OO with unknown dissimilarity. In BamSOO, Wang et al. (2014) use GP-UCB to reduce the number of evaluations required when running SOO alone. By using SOO, they can in return reduce the optimization costs of the acquisition function. Gupta et al. (2021) improve upon the basic version of BamSOO by a more elaborate partitioning scheme: Instead of dividing a cell into $k$ children along the longest side of the cell, they divide along the $b$ longest dimensions into $a$ cells, where $b^a = k$. Salgia et al. (2021) use a random walk based strategy on a tree to improve over the grid-based optimization of the GP-UCB acquisition function. For the Matérn and Squared Exponential kernel, they achieve order optimal regret, but the computational complexity is $\mathcal{O}(N^4)$.

**Work with the canonical pseudo-metric.** Shekhar & Javidi (2018) use the GP's canonical pseudo-metric to improve the numerical optimization by pruning the search regions. They additionally keep track of the posterior to only evaluate at locations where posterior uncertainty exceeds the cell's upper bound. Rando et al. (2022) follow this approach and additionally introduce a Nyström approximation, which allows for approximate inference in $\mathcal{O}(N^2 d_{\text{eff}}^2)$, where $d_{\text{eff}}$ is the effective dimension of the search space. Contal et al. (2015) replace the GP-UCB bounds with bounds derived from the pseudo-metric, but do not use a hierarchical approach, i.e. they construct bounds for individual points. They update the posterior and the exploration terms after every new observation.

Grill et al. (2018) apply the optimistic optimization principle to a one-dimensional Brownian walk. A minor difference is, that they evaluate a cell at the corners of an interval and not in the center. There are cases, where this is advantageous, e.g. think of samples from a GP with a polynomial or linear kernel. However, the number of corners scales exponentially with the dimension.

### 4.2 Work on scalable BO

TurBO (Eriksson et al., 2019) uses independent local GP models for a number of trust regions. Trust regions are shrinked or expanded based on heuristics capturing how much progress was made in the previous steps. A global Bandit strategy is used to decide in which of the trust regions to continue the search. Due to the heuristics involved it is less amendable for theoretical analysis. Other approaches to speed up BO rely on Bayesian neural networks (Snoek et al., 2015), lower dimensional embeddings (Wang et al., 2016), approximations to the GP (e.g. Jimenez & Katzfuss, 2022) or additive model assumptions (Han et al., 2021; Mutny & Krause, 2018).

## 5 Regret

While computational and not sample efficiency is our main motivation to apply the OO principle in the BO setting, we show that the resulting method nevertheless leads to non-trivial regret. In particular, it is asymptotically regret-free in the limit $\lim_{N \to \infty} R_N / N$. Here, $R_N$ denotes the cumulative regret defined as $R_N = \sum_{n=0}^{N} f(x^*) - f(x_n)$.

Building upon arguments from Munos (2011) and Shekhar & Javidi (2018), we obtain the following guarantee for the cumulative regret:

**Proposition 1** *Let $\mathcal{X}$ be finite, $\epsilon \in (0,1)$ and $\beta_n = 2 \log(2N|\mathcal{X}_n|/\epsilon)$. Running GP-OO with $\beta_n$ for a sample $f$ from a GP with mean function zero and covariance function $k(x,x)$, the following regret bound holds with*

*probability* $1 - \epsilon$:

$$R_N \leq \beta^{1/2} \sum_{n=1}^{N} \Delta(\log n)$$

*where* $\beta = max\{\beta_1, ..., \beta_N\}$ *and* $\Delta(n)$ *denotes the diameter of a cell evaluated at depth n.*

A full proof is provided in the Supplements. The high-level idea is that either the explored cell $\mathcal{X}_n$ contains the optimum $x^*$, then the simple regret $|f(x^*) - f(x)|$ is trivially upper bounded by the maximal deviation $\beta^{1/2}\Delta(\mathcal{X}_n)$. Or the cell does not contain the optimum, but then then its utility was higher than the one of a node containing the optimum in its region, and thereby higher than the optimum itself. For the cumulative regret we assume the worst-case of a uniformly growing tree. For a broad class of kernels, the bound in Proposition 1 can be further specified:

**Proposition 2** *Assume the GP's canonical pseudo-metric d satisfies* $d(x,y) = C\|x-y\|_2^\alpha$, *where* $C > 0, \alpha > 0, m/\alpha > 1$. *Running GP-OO on a finite domain* $\mathcal{X} \subset [0,1]^m$ *with regular partitions, one has a worst-case cumulative regret* $R_N$ *of*

$$\tilde{\mathcal{O}}(N^{1-\alpha/m}(\log N)^{\alpha/m})$$

*with high probability The* $\tilde{\mathcal{O}}(\cdot)$ *notation supresses poly-logarithmic factors.*

In particular, for the squared exponential kernel and Matérn kernels with half-integer values $\nu \geq 3/2$, one has $\alpha = 1$ and $R_N \in \tilde{\mathcal{O}}(N^{1-1/m}(\log N)^{1/m})$. For comparison, the regret in GP-UCB grows as $\tilde{\mathcal{O}}(\sqrt{N}\log(N)^{\frac{m+1}{2}})$ for the squared exponential kernels and thereby scales better to higher dimensions. For the Matérn class, GP-UCB is guaranteed to have regret at most $\tilde{\mathcal{O}}(N^{\frac{\nu+m(m+1)}{2\nu+m(m+1)}})$ for $\nu \geq 1$. Our bound is tighter, e.g., for the values $\nu = 3/2$ or $\nu = 5/2$ often used in practice.

Shekhar & Javidi (2018) establish regret bounds in terms of the near-optimality dimension $\tilde{m}$. This measure is commonly used in optimistic optimization to characterize the size of the set of $\epsilon$-optimal points in terms of packing numbers. The near-optimality dimension does not only depend on the underlying metric, but also on the function $f$ itself and is thereby a random variable. Smaller values are associated with deeper growing trees, whereas larger values lead to more balanced, uniformly growing trees. Assuming the worst case of $\tilde{m} = m$, the bounds in Shekhar & Javidi (2018) become $\tilde{\mathcal{O}}(N^{1-\alpha/m})$ for the Matérn class and match our worst-case bound. However, we restricted the analysis to finite domains $\mathcal{X}$, and $|\mathcal{X}|$ enters our bound logarithmically.

Grill et al. (2018) showed that in the specific case of Brownian motion, the tree built during optimistic optimization does not grow with the worst-case uniform rate.

## 6 Experiments

We empirically compare GP-OO to GP-UCB, EI, TurBO and DiRect in terms of regret and time, on synthetic, 3d samples from GPs, and Benchmark functions from Surjanovic & Bingham (2022). All experiments were performed on a desktop machine. Since computational efficiency is central to our analysis, we also report wall-clock runtimes. For GP-UCB and EI, we use implementations from Emukit (Paleyes et al., 2019; Gardner et al., 2018). For TurBO, we build upon the published code accesible at `https://github.com/uber-research/TuRBO`. However, we turn off the optimization of the lengthscale an noise scale and instead provide it with ground-truth values, for a fair comparison with the other methods that also have this information. Also, we changed TurBO's default batch size from 10 to 2 to make it the same as GP-OOs. Other than that, we use its default settings. For DiRect, we use the python implementation available at `https://github.com/swyoon/DiRect/blob/master/LICENSE`.

### 6.1 Experiments on synthetic functions

**Regular partitions.** We begin with on-model experiments for a common setting in Bayesian optimization, where GP-OO can showcase its speed advantages: 20 samples from a GP with squared exponential kernel

(lengthscale $l = 0.2$) on the unit cube $\mathcal{X} = [0,1]^3$. For all experiments, the noise level for GP-UCB is set to a very small constant $\lambda = 0.0005$ since we assume noiseless observations. For GP-OO, we use $\beta_n = 2\log(2|\hat{\mathcal{X}}_n|/\epsilon)$ with a discretization of $1/l$ points along each dimension. For additional heuristical choices of $\beta$ as well as constant choices of $\beta_n \in \{0.1, 1, 10, 100\}$, see Appendix. For GP-UCB, we use $\beta_n = 2\log(2|\hat{\mathcal{X}}|n^2\pi^2/6\epsilon)$ according to theory with a grid search over the size of the discretization $|\hat{\mathcal{X}}|$ with values in $\{1, 10, 100, 1000\}$, as well as constant values $\beta$ in $\{1, 10, 100, 1000\}$. For EI, we did a grid search over the jitter parameter with values in $\{0.0001, 0.001, 0.01, 0.1, 1\}$. GP-OO and GP-UCB require the specification of a confidence level $\epsilon$ that is set to 0.05 for both methods. Figure 3 shows the simple regret $min_n r_n$ and the average cumulative regret $R_n/n$ in terms of number of function evaluations $n$. Figure 4 displays the runtime of GP-UCB, GP-OO and TurBO. In terms of number of function evaluations there is no improvement over the state-of-the art, however in terms of runtime GP-OO is orders of magnitude faster than classical BO approaches and also improves over TurBO's runtime. In terms of sample-efficiency, GP-OO improved over DiRect. This indicates that GP-OO can exploit the smoothness encoded in the prior. For a more detailed comparison of DiRect and GP-OO, see also Appendix A.3.

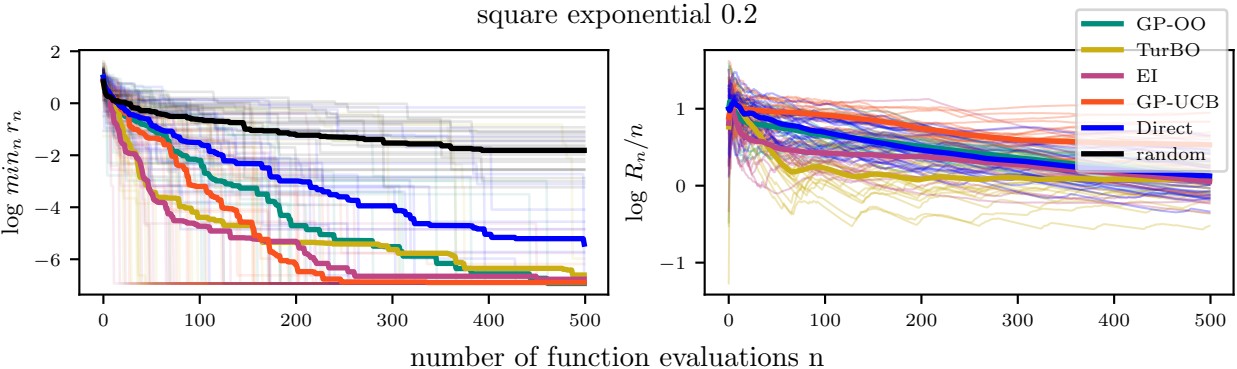

Figure 3: Best simple regret (left) and average cumulative regret (right) on synthetic functions sampled from a GP.

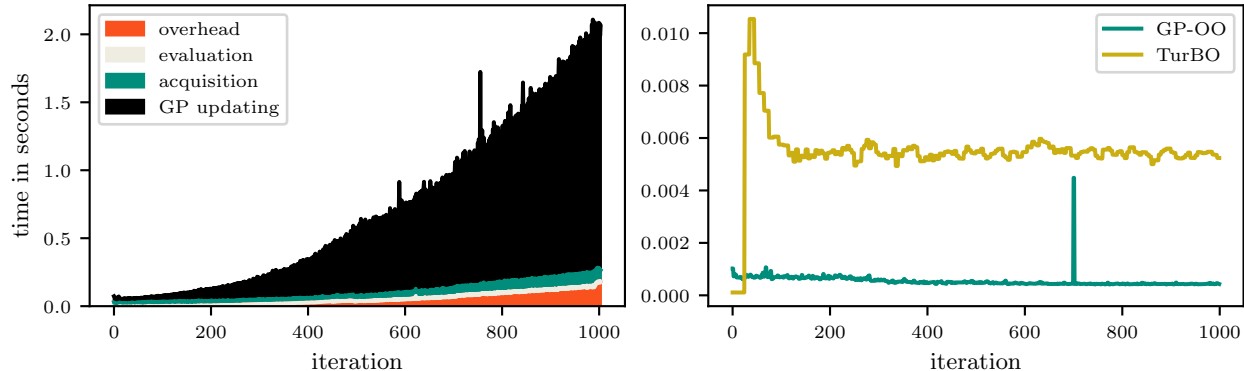

Figure 4: Runtime per iteration [seconds] for on-model optimization of GP-samples. Left: Time for different components of GP-UCB iterations. Right: Total time per iteration (averaged over batch size) for GP-OO and TurBO.

**Non-regular partitions.** For demonstration, we also explore a setting with non-regular optimal partitions, where GP-OO can not perform as well. We sample 100 functions from a GP with quadratic kernel $k(x,y) = (x^T y)^2$ with bias 0 on the domain $[-1,1]^3$. We consider GP-UCB and GP-OO, once with euclidean partitions and once with partitions optimized with respect to the canonical metric. The exploration constant was set to $\beta = 1$. The results shown (Figure 5) show that it is advantageous to optimize the partitioning scheme with respect to the canonical metric. However, we restrict ourselves to partitions with axis-aligned cells, i.e. the perfectly correlated corner points with $k(x,y) = 1$ and $d(x,y) = 0$ end up in different cells. At this

point, GP-UCB has a clear advantage, as information can be shared across the search space between the corner points. Also, the run-time advantage decreases due to the numerical optimization of the partitions.

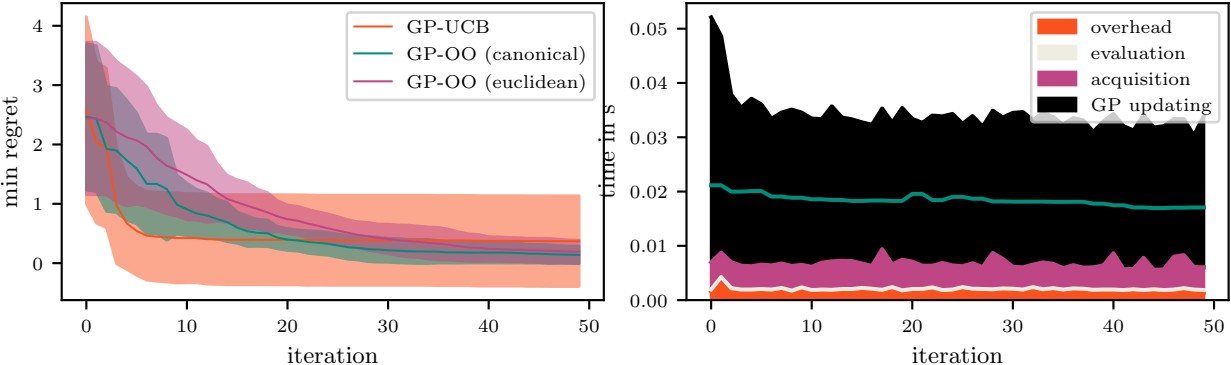

Figure 5: Experiment with a quadratic kernel. Right: Wall-clock time with GP-UCB. Green line: Wall-clock time with GP-OO.

## 6.2 Experiments on benchmarks

We consider the minimization of common optimization benchmark functions, on domains with dimensions ranging from 2 to 10. Since the functions are deterministic, and GP-OO is deterministic up to random tie breaks, we randomly sample 10 domains $\mathcal{X}$ to obtain variation (see Supplements). Since evaluations of the benchmark functions are very cheap, we artificially simulated higher function evaluation costs by adding post-hoc costs from 0.001 to 1 Second to the evaluation times. We use a Matérn kernel with $\nu = 3/2$ and length-scales as recommended by Rando et al. (2022), who obtained them with a grid search. For the exploration constant $\beta$ we did a grid search over $\{0.1, 1, 10, 100\}$ for GP-UCB and GP-OO. For EI, we did a grid search for the jitter parameter with values as above. We use confidence level $\epsilon = 0.05$. Figure 9 shows the minimal simple regret over the number of iterations and Figure 7 shows the minimal simple regret over time. The Supplements contain additional figures for function evaluation costs of 0.01 and 0.1 Seconds, as well as a figure showing cumulative run-times.

GP-OO is competitve with the BO based approaches on 7 of the 12 benchmarks (Dixon-Price, Bohachevsky A, B and C, Ackely, Hartmann and Shekel. This even holds in number of function evaluations and not only wall-clock time. However, we do not find a consistent advantage over DiRect and in some cases GP-OO is outperformed by Direct. Since the benchmark functions are typically not within the RKHS, it cannnot be guranteed that GP-OO converges to the global optima. In particular on Rosenbrock and Beale, it happened that GP-OO got stuck in very bad local optimal. For DiRect this happend less, indicating that it was harmful to rely on the smoothness assumptions, when they are not fully met. TurBO is also less prone to get stuck in local optima as it has an in built criterion for random restarts of trust regions once they converged to a (local) optimum.

## 7 Conclusion

BO and OO are closely connected through a mapping from the kernel-function to the canonical pseudo-metric. We showed that, for some kernels, this connection can be exploited to derive a computationally efficient method for the BO setting that captures prior information. For common choices of kernels, like the Matérn class, we outperform classical BO approaches, like GP-UCB and EI, if the objective function has low evaluation cost. In many cases, this simple approach achieves performance similar to scalable BO approaches highly optimized for runtime. On the other hand, the strong reliance on the prior assumptions without being able to adapt the hyperparameters of the prior on the fly, is currently still a limitation in practical settings. A setting, where one quickly wants to find the maximum of synthetic GP sample, might already be interesting application cases of GP-OO: Grill et al. (2018) suggest to use Optimistic Optimization as a subroutine for Thompson sampling. Instead of sampling the function values from the posterior over an

entire grid and then taking its maximum, one runs optimistic optimization and only samples the function values at the evaluation locations found with optimistic optimization. In this setting, the hyperparameters of the posterior are known. The same goes for BO algorithms like Maximum Entropy Search or Maximum Value Entropy Search, where, one has to generate distributions of the (location of the) maximum from "on-model "samples from the posterior. Nonetheless, our work shows that the ability to use prior information in Bayesian optimization does not *have* to be expensive per se. Of particular conceptual importance is the insight that optimization can be performed *without* explicitly tracking a posterior.

**Acknowledgments**

This work was supported by Microsoft Research through its PhD Scholarship Programme. The authors thank the International Max Planck Research School for Intelligent Systems (IMPRS-IS) for supporting Julia Grosse by non-financial means. Philipp Hennig and Julia Grosse gratefully acknowledge financial support by the European Research Council through ERC StG Action 757275 / PANAMA; the DFG Cluster of Excellence "Machine Learning - New Perspectives for Science", EXC 2064/1, project number 390727645; the German Federal Ministry of Education and Research (BMBF) through the Tübingen AI Center (FKZ: 01IS18039A); and funds from the Ministry of Science, Research and Arts of the State of Baden-Württemberg.

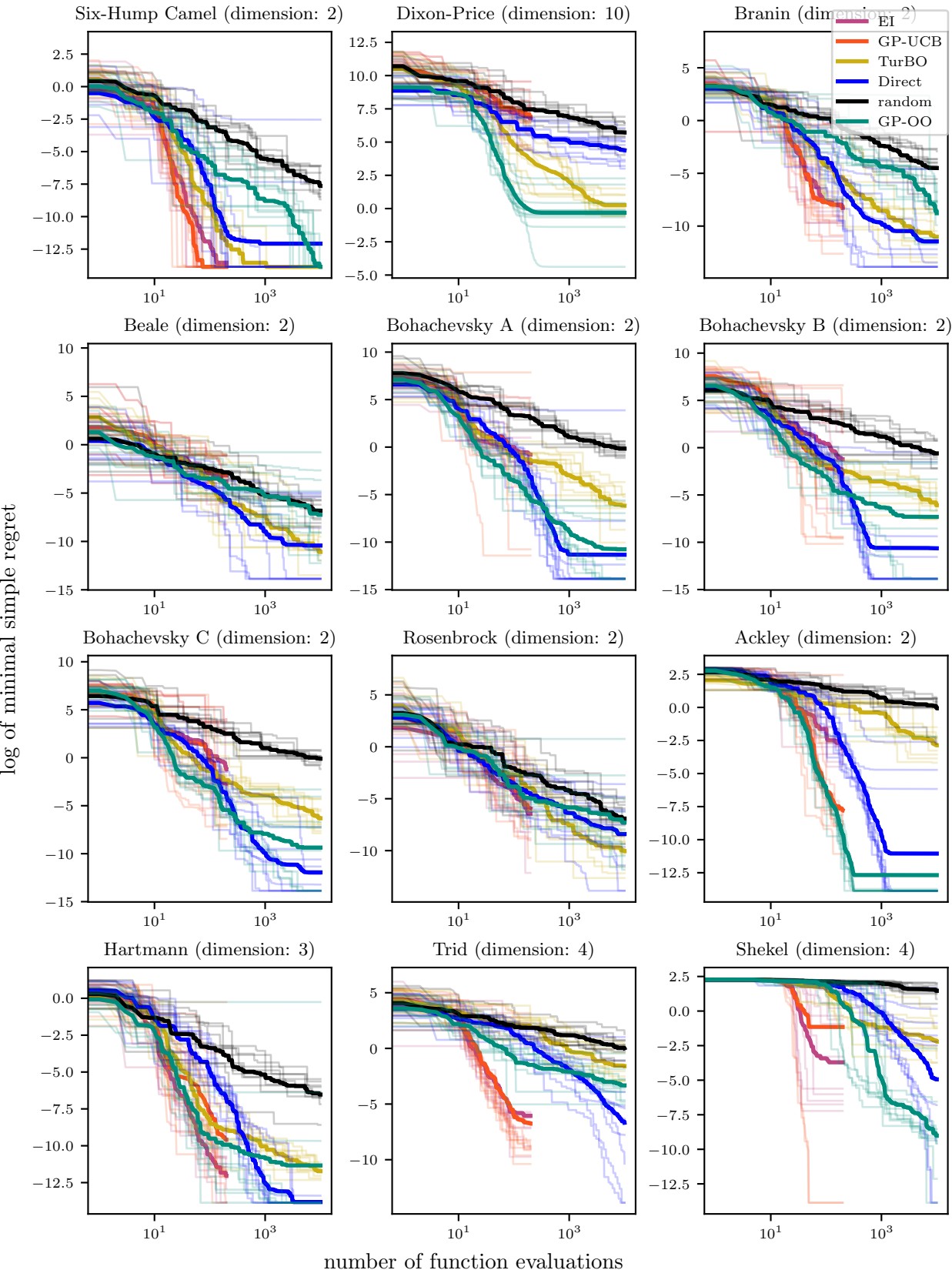

Figure 6: Optimization performance (minimal function values found) in terms of number of iterations on common benchmarks with GP-OO, GO-UCB, EI, TurBO, DiRect, and random selection of evaluations

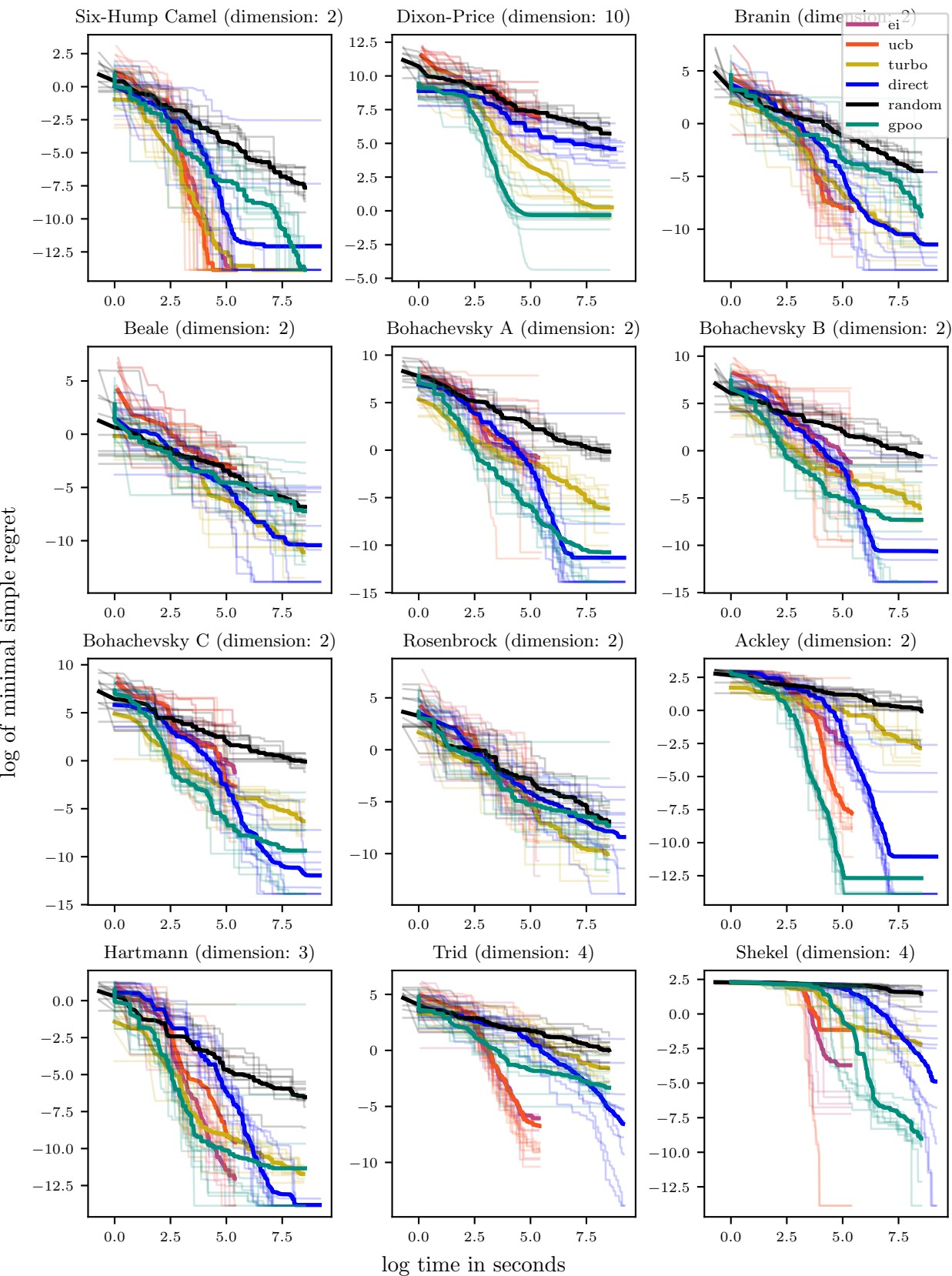

Figure 7: Optimization performance (minimal function values found) in terms of wall clock time for simulated function evalution times of 1 Second on common benchmarks with GP-OO, GO-UCB, EI, TurBO, DiRect and random selection of evaluations.

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

## A  Appendix

### A.1  Theoretical Analysis

#### A.1.1  Background

The following three facts will be used during the analysis:

1. **Deviation inequality** For a sample $f \sim \mathcal{G}(0, k)$ from a centered GP with kernel function $k$, it holds

$$\forall u, \mathbb{P}(|f(x) - f(y)| \geq u) \leq 2 \exp\left(-\frac{u^2}{2d(x, y)^2}\right), \tag{10}$$

   where $d(x, y) = \sqrt{k(x, x) + k(y, y) - 2k(x, y)}$.

2. **Hölder's inequality** Let $a_1, ..., a_N, b_1, ..., b_N$ be real numbers.

$$\sum_{n=1}^{N} |a_n b_n| \leq \left(\sum_{n=1}^{N} |a_n|^p\right)^{1/p} \left(\sum_{n=1}^{N} |b_k|^q\right)^{1/q} \tag{11}$$

   with $p, q \geq 1$ and $\frac{1}{p} + \frac{1}{q} = 1$

3. **Growth of Harmonic numbers** The $N$-th Harmonic number $H_N = \sum_{n=1}^{N} \frac{1}{n}$ grows logarithmically in $N$:

$$H_N \in \Theta(\log N) \tag{12}$$

#### A.1.2  Upper bounds on the supremum of a cell

**Lemma 1.** Assume $\mathcal{X}$ is finite and $f \sim \mathcal{GP}(0, k)$. Pick $\epsilon \in (0, 1)$ and set $\beta_n = 2 \log(2|\mathcal{X}_n|N/\epsilon)$. Then

$$\forall n \sup_{x \in \mathcal{X}_n} |f(x) - f(x_n)| \leq \beta_n^{1/2} \Delta(\mathcal{X}_n)$$

holds with probability $\geq 1 - \epsilon$. $\mathcal{X}_n$ is the cell visited at step $n$ with center $x_n$ and $\Delta(\mathcal{X}_n) = \sup_{x \in \mathcal{X}_n} d(x, x_n)$.

**Proof of Lemma 1.** Fix $n$. Applying a union bound and 10, one obtains for all $u_n$

$$\mathbb{P}(sup_{x \in \mathcal{X}_n} |f(x) - f(x_n)| \geq u_n) \tag{13}$$

$$\leq \sum_{x \in \mathcal{X}_n} \mathbb{P}(|f(x) - f(x_n)| \geq u_n) \tag{14}$$

$$\leq 2 \sum_{x \in \mathcal{X}_n} \exp\left(-\frac{u_n^2}{2d(x, x_n)^2}\right) \tag{15}$$

$$\leq 2|\mathcal{X}_n| \exp\left(-\frac{u_n^2}{2\Delta(\mathcal{X}_n)^2}\right) \tag{16}$$

For $u_n = \beta_n^{1/2} \Delta(\mathcal{X}_n)$, it holds with probability $1 - \epsilon/N$, that $sup_{x \in \mathcal{X}_n} |f(x) - f(x_n))| \leq \beta_n^{1/2} \Delta(\mathcal{X}_n)$. Taking another union bound over $N$ the statement holds.

#### A.1.3  Upper bound on the regret

**Lemma 2.** Running GP-OO with $\beta_n$ as specified in Lemma 1 and the canonical pseudo-metric $d$, the simple regret $r_n = f^* - f_n$ is bounded by $\beta_n^{1/2} \Delta(\mathcal{X}_n)$ for all $n$ with probability $1 - \epsilon$.

**Proof of Lemma 2.** This statement can be shown with typical arguments from the literature on optimistic optimization. We consider the cases $x^* \in \mathcal{X}_n$ and $x^* \notin \mathcal{X}_n$ separately.

Case 1: $x^* \in \mathcal{X}_n$

$$r_n = f^* - f_n = sup_{x \in \mathcal{X}_n} |f(x) - f(x_n)| \leq \beta_n^{1/n} \Delta(\mathcal{X}_n) \tag{17}$$

by Lemma 1.

Case 2: $x^* \notin \mathcal{X}_n$.
Because $x_n$ was explored nevertheless, there is a node $x_{n'}$ on the optimal path with $x^* \in \mathcal{X}_{n'}$, that was explored at step $n' < n$, such that

$$f(x_n) + \beta_n^{1/2} \Delta(\mathcal{X}_n) \geq f(x_{n'}) + \beta_{n'}^{1/2} \Delta(\mathcal{X}_{n'}) \tag{18}$$

Then, $f(x_{n'}) - f(x_n) \leq \beta_n^{1/2} \Delta(\mathcal{X}_n) - \beta_{n'}^{1/2} \Delta(\mathcal{X}_{n'})$. For the regret one obtains in combination with Lemma 1:

$$r_n = f^* - f_n \tag{19}$$
$$\leq \left[ f(x^*) - f(x_{n'}) \right] + \left[ f(x_{n'}) - f(x_n) \right] \tag{20}$$
$$\leq \left[ \beta_{n'}^{1/2} \Delta(\mathcal{X}_n') \right] + \left[ \beta_n^{1/2} \Delta(\mathcal{X}_n) - \beta_{n'}^{1/2} \Delta(\mathcal{X}_{n'}) \right] \tag{21}$$
$$= \beta_n^{1/2} \Delta(\mathcal{X}_n) \tag{22}$$

**Proposition 1.** Assume $\mathcal{X}$ is finite and $f \sim \mathcal{GP}(0, k)$. Pick $\epsilon \in (0, 1)$ and set $\beta_n = 2 \log(2|\mathcal{X}_n|N/\epsilon)$. For GP-OO with $k = 2$ children nodes, we obtain the following bound on the cumulative regret

$$\mathbb{P}\left( R_N \leq \beta^{1/2} \sum_{n=1}^{N} \Delta(\lfloor \log n \rfloor) \right) \geq 1 - \epsilon \tag{23}$$

where $\Delta(h)$ is the radius of a cell at depth $h$ and $\beta = \max\{\beta_1, ..., \beta_N\}$.

**Proof Proposition 1.** Simple consequence from Lemma 2, where we assume the worst case of a uniformly growing tree. The depth of a node after $n$ steps is at least $\lfloor \log(n) \rfloor$ in a uniformly growing tree.

### A.1.4   Bounds for common kernels

The following analysis is restricted to GP's, where the canonical pseudo-metric $d$ satisfies:

**Assumption 1.** There exist $C > 0, \alpha > 0$ such that $d(x, y) \leq C \|x - y\|_2^\alpha$, where $\| \cdot \|_2$ is the Euclidean norm. We additionally require $m/\alpha > 1$, where $m$ is the dimension of the domain.

According to Shekhar & Javidi (2018) the first part of Assumption 1 holds for the squared exponential kernel with $C = \sqrt{2/l}, \alpha = 1$ and the Matern kernels with half integer values. For $\nu = 1/2$, one has $\alpha = 1/2$ and for all other half-integer values $\alpha = 1$.

**Lemma 3.** [Bubeck et al. (2011)] Assume that $\mathcal{X}$ is a $m$-dimensional hypercube $[0, 1]^m$ and consider the dissimilarity $d(x, y) = C\|x - y\|_2^\alpha$, where $C > 0, \alpha > 0$. Define the partitioning by recursively splitting the hypercube in the middle along its longest side (ties broken arbitrarily). One has

$$\Delta(\mathcal{X}_n) \leq diam(\mathcal{X}_h) \leq C(2\sqrt{m})^\alpha \left( \frac{1}{2^{\alpha/m}} \right)^h$$

for the cell $\mathcal{X}_h$ of a node at depth $h$.

**Proof of Lemma 3.** See Example 1 in Bubeck et al. (2011).

**Proposition 2** *Assume the GP's canonical pseudo-metric $d$ satisfies $d(x, y) = C\|x - y\|_2^\alpha$, where $C > 0, \alpha > 0, m/\alpha > 1$. Running GP-OO on a finite domain $\mathcal{X} \subset [0, 1]^m$ with regular partitions, one has a worst-case cumulative regret $R_N$ of*

$$\tilde{\mathcal{O}}(N^{1-\alpha/m}(\log N)^{\alpha/m})$$

*with high probability. The $\tilde{\mathcal{O}}(\cdot)$ notation supresses poly-logarithmic factors.*

**Proof of Proposition 2** It follows from Proposition 1 that $R_N \in \tilde{\mathcal{O}}(\sum_{n=1}^{N} \Delta(\lfloor \log n \rfloor))$ with probability $1 - \epsilon$. It remains to bound $\sum_{n=1}^{N} \Delta(\lfloor \log n \rfloor)$. For Equation (24), we use Lemma 3 and for Equation (29), we apply Hölder's inequality (11) with $q = m/\alpha$ and $p = \frac{1}{1-\alpha/m}$.

$$\sum_{n=1}^{N} \Delta(\lfloor \log n \rfloor) \leq C\left(2\sqrt{m}\right)^\alpha \sum_{n=1}^{N} \left(\frac{1}{2^{\alpha/m}}\right)^{\lfloor \log n \rfloor} \tag{24}$$

$$\leq C(2\sqrt{m})^\alpha \sum_{n=1}^{N} \left(\frac{1}{2^{\alpha/m}}\right)^{\log n - 1} \tag{25}$$

$$= C(2\sqrt{m})^\alpha 2^{\alpha/m} \sum_{n=1}^{N} \left(\frac{1}{2^{\alpha/m}}\right)^{\log n} \tag{26}$$

$$= C_1 \sum_{n=1}^{N} \left(\frac{1}{2^{\alpha/m}}\right)^{\log n} \tag{27}$$

$$= C_1 \sum_{n=1}^{N} \left(\frac{1}{n^{\alpha/m}}\right) \tag{28}$$

$$\leq C_1 \left(\sum_{n=1}^{N} 1\right)^{1-\alpha/m} \left(\sum_{n=1}^{N} \left(\frac{1}{n^{\alpha/m}}\right)^{m/\alpha}\right)^{\alpha/m} \tag{29}$$

$$= C_1 N^{1-\alpha/m} H_N^{\alpha/m} \tag{30}$$

where $C_1 = C(2\sqrt{m})^\alpha 2^{\alpha/m}$ and $H_N$ being the $N$-th harmonic number. Together with (12), this implies:

$$\sum_{n=1}^{N} \Delta(\lfloor \log n \rfloor) \in \mathcal{O}(N^{1-\alpha/m}(\log N)^{\alpha/m}) \tag{31}$$

## A.2 Additional experimental details

The experiments were implemented in Python 3.9.1. and run on a machine with macOS 12.3.1, a 4 GHz Quad-Core Intel Core i7 CPU and 32 GB RAM or on a machine with macOS 12.2.1, a 2,7 GHz Dual-Core Intel Core i5 CPU and 16 GB RAM in the case of the Experiment in Section 6.1.

### A.2.1 Experiment with benchmark functions

Table 1 lists the domains and hyperparameters used in the experiment with the benchmark functions. We ran TurBO in its default configuration with 5 trust regions and a batch size of 10. For each run, we sampled a sub-domain by choosing uniformly at random new lower and upper boundaries for the intervals along each dimension, such that the new boundaries are inbetween the previous ones and the location of the minimum. In this way, the location of the minimum stays the same as in the original domain.

Figure 9 shows the minimal function value found for the benchmark functions and Figure 10 the cumulative runtime. While in terms of number of function evaluations GP-OO does not improve over existing methods, it is the fastest.

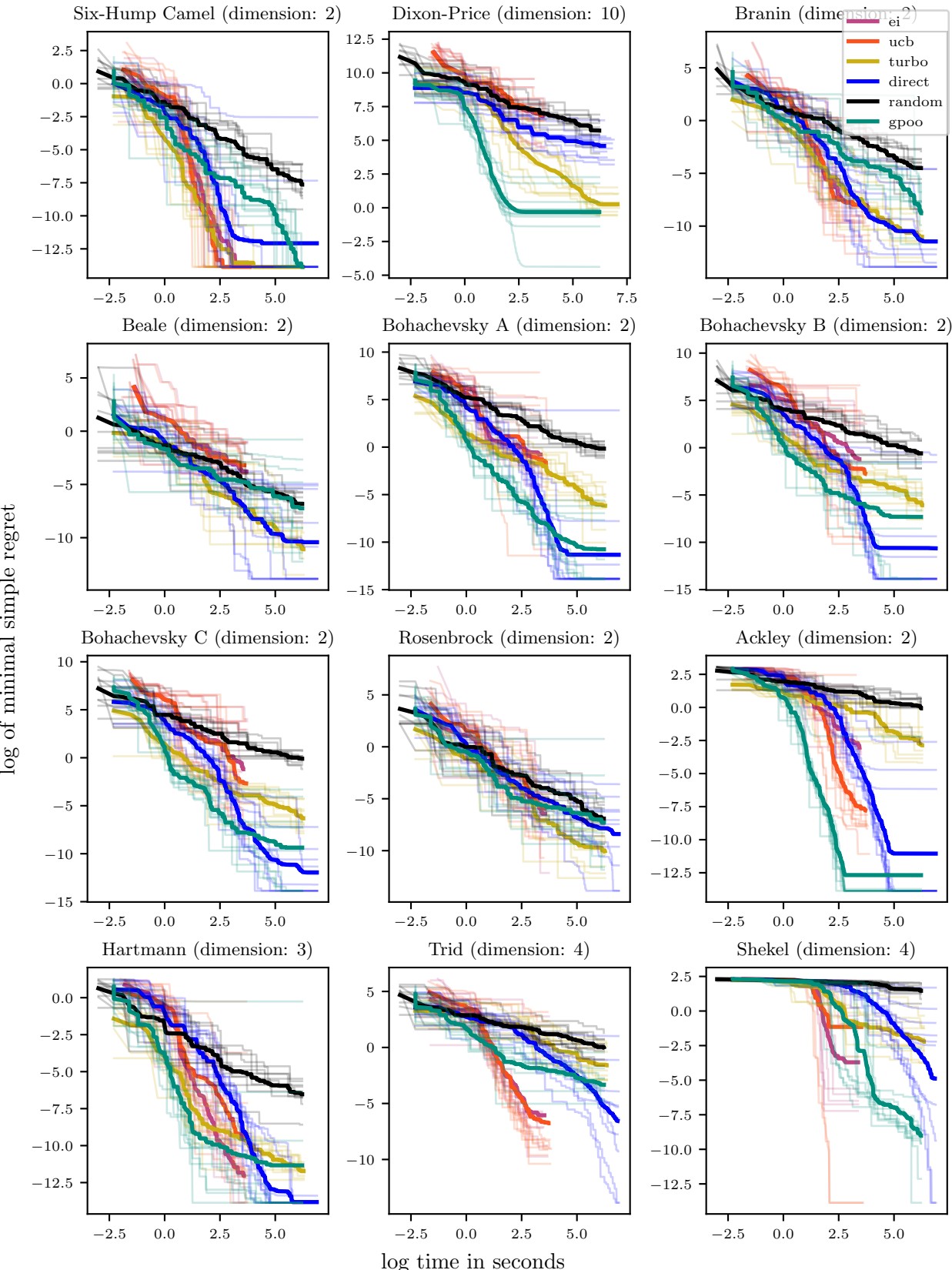

Figure 8: Minimal function values found on common benchmark datasetsfor simulated function evaluation costs of 0.1 Second.

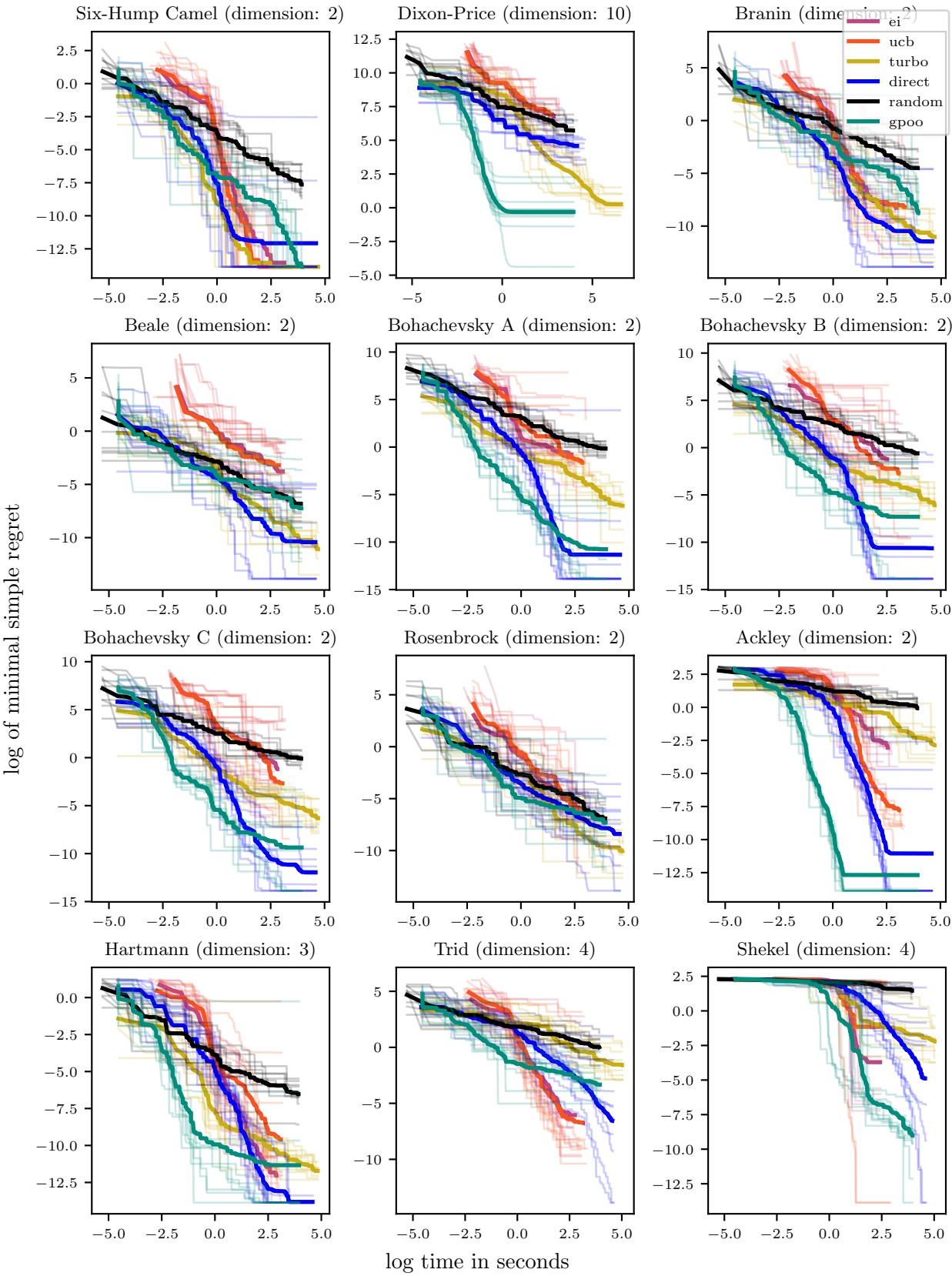

Figure 9: Minimal function values found on common benchmark datasets for simulated function evaluation costs of 0.01 Second.

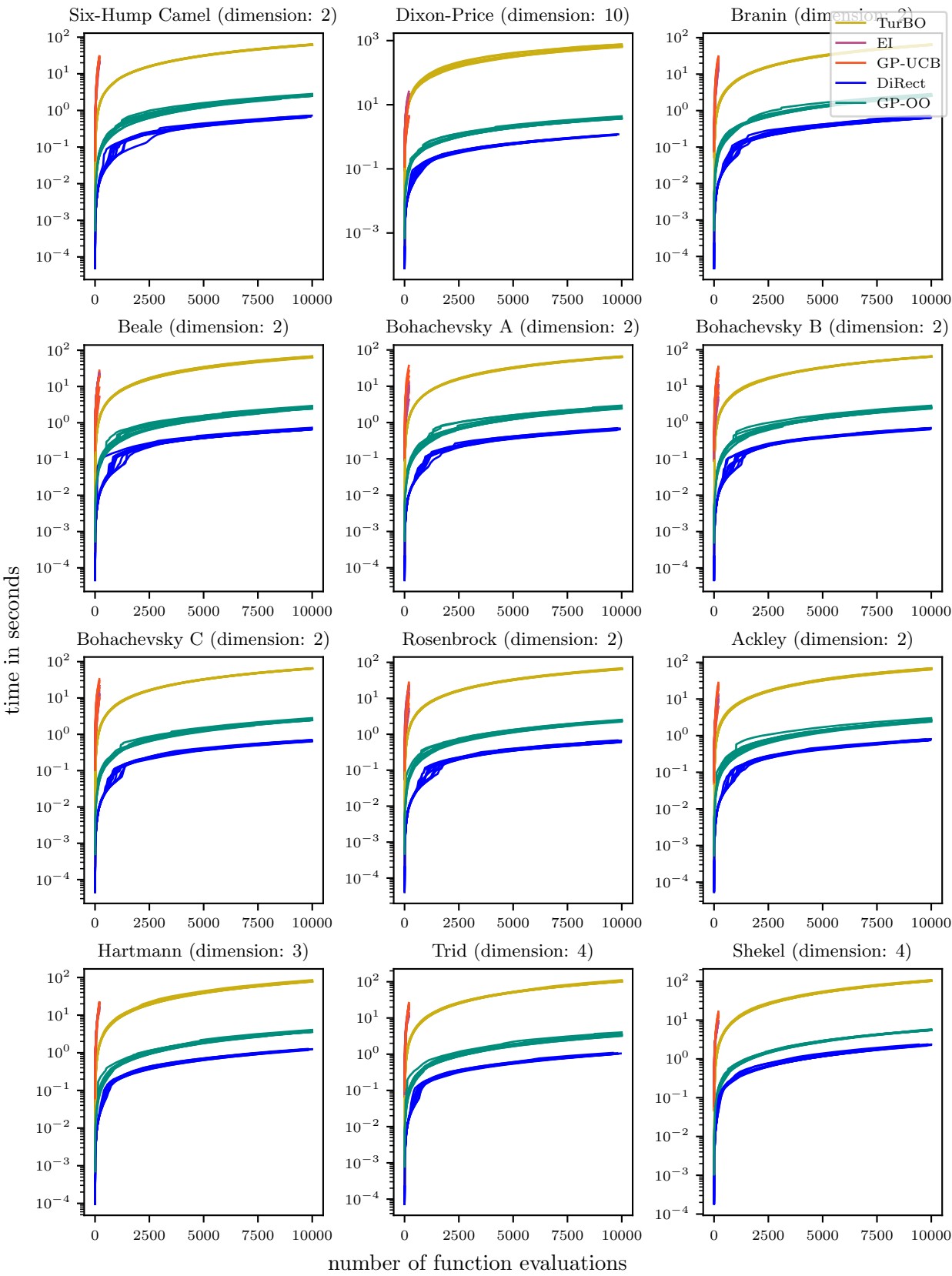

Figure 10: Cumulative runtimes for common benchmark datasets.

Table 1: Hyperparameters for experiment with benchmark functions

| Benchmark | Domain | Lengthscale $l$ | $\beta$ (GP-UCB) | $\beta$ (GP-OO) | jitter (EI) |
|---|---|---|---|---|---|
| Branin | $[-15, 15]^2$ | 0.5 | 1 | 100 | 0.001 |
| Six-Hump-Camel | $[-2, 2]^2$ | 0.5 | 1 | 10 | 0.0001 |
| Beale | $[-4.5, 4.5]^2$ | 1 | 0.1 | 100 | 0.0001 |
| Bohachevsky a | $[-100, 100]^2$ | 1.7 | 0.1 | 10 | 1 |
| Bohachevsky b | $[-100, 100]^2$ | 1.7 | 1 | 10 | 0.1 |
| Bohachevsky c | $[-100, 100]^2$ | 1.7 | 0.1 | 10 | 0.0001 |
| Rosenbrock | $[-3, 3]^2$ | 0.7 | 1 | 100 | 0.0001 |
| Ackley | $[-35, 35]^2$ | 3.5 | 1 | 10 | 0.001 |
| Hartmann | $[0, 1]^3$ | 0.3 | 1 | 0.1 | 0.0001 |
| Trid | $[-16, 16]^4$ | 10.75 | 1 | 100 | 0.0001 |
| Shekel | $[0, 10]^4$ | 1.75 | 1 | 10 | 0.001 |
| Dixonprice | $[-10, 10]^{10}$ | 2 | 10 | 0.1 | 0.1 |

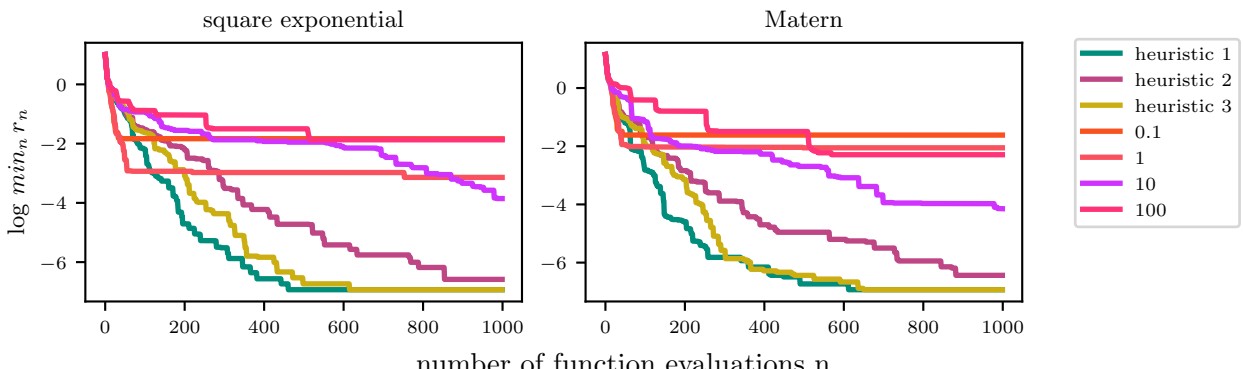

Figure 11: Performance of GP-OO for different choices of $\beta_n$ on 20 samples from a GP.

Table 2: Estimated probability that all upper bounds in a tree with 10 levels hold for samples from a GP.

|  | square exponential | Matérn |
|---|---|---|
| heuristic 1 | 15% | 0% |
| heuristic 2 | 83% | 94% |
| heuristic 3 | 14% | 7% |

### A.2.2  Heuristical choice of $\beta$

Figure 11 shows GP-OO's performance in the experiment on synthetic samples from a GP as described in Section 6.1 for three heuristical choices of $\beta_n$ and constant values of $\beta_n \in \{0.1, 1, 100, 1000\}$. We performed the same analysis for a Matérn kernel with $\nu = 3/2$. According to theory, the upper bounds hold for $\beta_n = 2\log(2|\mathcal{X}_n|N/\epsilon)$ with probability $1 - \epsilon$, but typically a full union bound over both the size of the domain and the total number of observations is too loose. The heuristics are inspired by the observation that the number of "effectively independent" points in the domain is $1/l$ along each dimension. In the following, $l$ denotes the lenghtscale and $d$ the dimension of the search domain. $C$ is a constant that we set to 1 for the square exponential kernel and $3/2$ for the Matérn kernel.

- Heuristic 1: $N = 1$ and $\hat{\mathcal{X}}_n = (1/l)^d$

- Heuristic 2: $N = (C/l)^d$ and $\hat{\mathcal{X}}_n = (C/l)^d$

- Heuristic 3: $N = 1000$ (as we run the search for a total of 1000 iterations) and $\hat{\mathcal{X}}_n = (1/l)^d$

In addition, we empirically estimated the probability with which our heuristical choices for $\beta$ result in valid upper bounds. To do so, we sampled 100 functions from a GP and for each sample build a fully balanced search tree up to depth 10. For each sampled function and each node, we checked if the upper-bound holds for the corresponding cell. The following table shows the percentage of samples for which *all* upper-bounds from depth 0 to depth 10 did hold. All functions were sampled on a $25 \times 25 \times 25$ grid with a square exponential and Matérn kernel with lengthscale 0.2. $\epsilon$ is set 0.05, so one would expect 95% of the bounds to hold under a union bound approach. For the results, see Table 2. Higher percentages indicate the $\beta$ is too conservative and lower percentages indicate that the optimization might be too greedy. We find that, even though the latter is the case for our heuristics, this does not immediatly lead to decreased search performance as shown by Figure 11.

### A.3  Ablation: Does encoding the smoothness help?

We compare GP-OO to DiRect (Jones et al., 1993), a global optimization algorithm that is also based on iterative refinement of axis-aligned rectangles. DiRect assumes that the function fulfills a Lipschitz condi-

Table 3: Discretization

| dimension/lengthscale | 3/1 | 3/0.1 | 3/0.05 | 2/0.5 | 2/0.05 | 2/0.005 | 1/0.5 | 1/0.05 | 1/0.005 |
|---|---|---|---|---|---|---|---|---|---|
| discretization per dimension | 40 | 40 | 40 | 100 | 200 | 500 | 1000 | 1000 | 2000 |

tion as well, however it does not allow to specify it. With the following experiment we investigate whether the Lipschitz assumption encoded in the prior allows GP-OO to improve over the performance of DiRect. To do so, we compare both methods on 20 samples from a GP prior with a Matérn kernel (Figure 12) or square exponential kernel (Figure 13) in different dimensions for different lengthscales. For GP-OO we set the exploration constant $\beta$ to according to heuristic 1 described in the previous section. The confidence parameter $\epsilon$ is set to 0.01. The domain is $[0,1]^d$ for dimension $d$. The functions sampled from the GPs are sampled over grids of different discretizations as specified in Table 3.

We find that GP-OO can exploit the smoothness in the prior, in particular on domains with smaller lengthscales, i.e. more local optimas. DiRect is known to converge slowly in the presence of many local optima as it sometimes spends a large number of function evaluations refining the grid around a local optimum before reaching the basin of a global optimum (Jones & Martins, 2021). Knowing the smoothness of the function might help GP-OO to reach a better trade-off here. For large lengthscales the advantage seems to disappear or even turns around on some samples. A potential reason could be a slight misspecification of the exploration constant beta. However, for this work the settings with small lengthscales are more relevant anyway, since in settings with large lengthscale BO is preferable as one does not need many evaluations there and BO than scales sufficiently well.

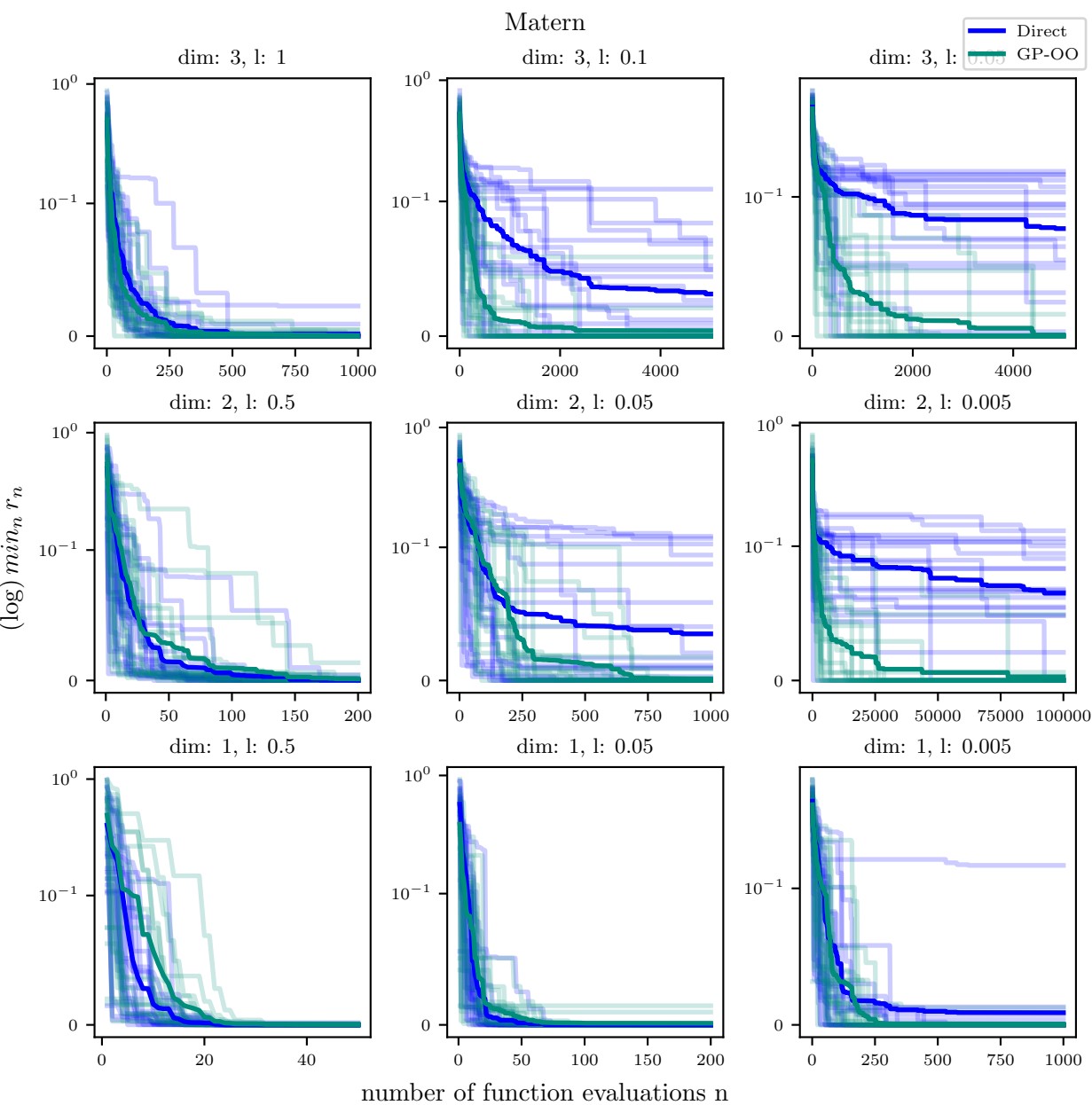

Figure 12: Performance of GP-OO and DiRect on 20 samples from a GP with a with Matérn kernel with different lengthscales and dimensions. The regret is normalized to $[0, 1]$ and the y-axis is semi-logarithmic with a linear threshold at 0.1.

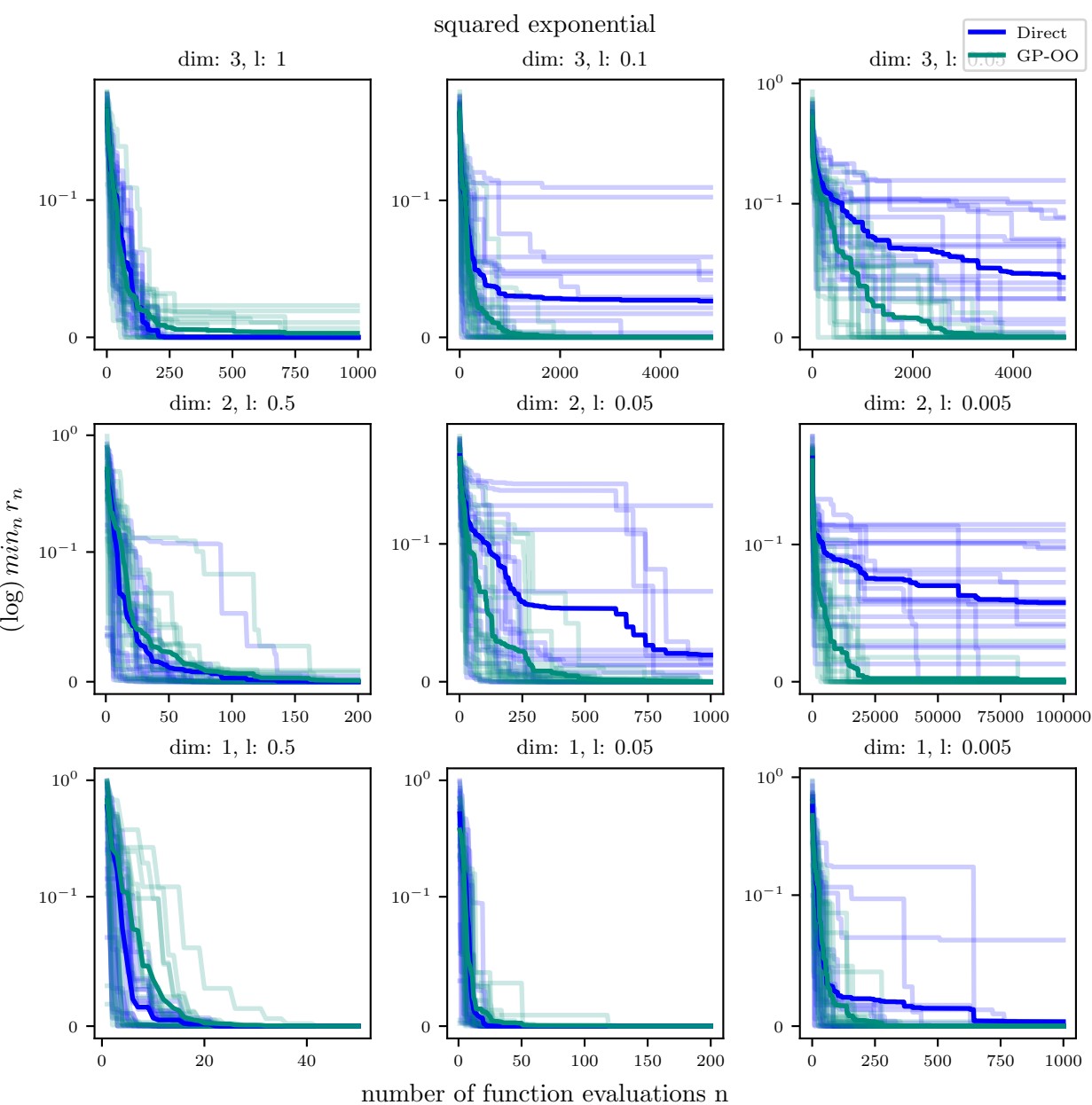

Figure 13: Performance of GP-OO and DiRect on 20 samples from a GP with a with square exponential kernel with different lengthscales and dimensions. The regret is normalized to $[0, 1]$ and the y-axis is semi-logarithmic with a linear threshold at 0.1.

