# OpenReview forum: "Optimistic Optimization of Gaussian Process Samples"
_TMLR — Accepted by TMLR_

### Review · Reviewer_U8Ye · 2023-06-22

**Summary Of Contributions:**

This paper proposes an optimization algorithm that bridges Bayesian optimization and optimistic optimization. In optimistic optimization, a pseudo-metric is used to define a binary search tree on which a UCB-like procedure is done, which results in a computational complexity of $O(N\log N)$. The authors propose to use a pseudo-metric based on a Gaussian process with known kernel (often used in Bayesian optimization with computational complexity $O(N^3)$), which does not increase the computational complexity. Theoretical analysis is done on the regret of the algorithm. Experiments are carried out on BO benchmarks showing that the algorithm is competitive with existing methods but has lower computational burden.

**Audience:**

Yes

**Broader Impact Concerns:**

As the paper is proposing a new optimization algorithm, I don't think there are any direct ethical implications. No broader impact statement is included.

**Claims And Evidence:**

Yes

**Requested Changes:**

In order of importance:
1. Include a wider range of baselines in the experiments, in particular those discussed in Section 4.1 that already mix Bayesian optimization and optimistic optimization.
2. It would strengthen the work to show results for dimensions in the hundreds (or maybe even thousands) as the proposed algorithm's cell splitting process is affected by the dimension.

**Strengths And Weaknesses:**

## Strengths
1. The paper is well-written and the contributions are presented clearly.
2. The proposed algorithm is mathematically motivated and theoretical analysis is done on the regret. The computational aspects of the algorithm are also thoroughly discussed.
3. The experimental results show that the algorithm is competitive with existing work but is less computationally heavy.

## Weaknesses
1. The baselines only include ones that rely primarily on GPs; however, the related work discusses other types of algorithms that are closer in spirit to the proposed algorithm.
2. The dimension of the objectives used in the experiments is at most 10, fairly small.
Please see the next field for further comments.

---

### Review · Reviewer_ZH9C · 2023-06-26

**Summary Of Contributions:**

The paper proposes a black-box optimization algorithm called GP-OO (Gaussian process optimistic optimization). The algorithm divides the search space into disjoint cells and iteratively selects a cell having the highest upper bound. The technical main contribution would be in the upper bound based search algorithm based on combining a local Lipschitz assumption and the GP assumption for the underlying objective function. The computation is quite simple based on which the authors mainly claim computational efficiency (not the sample efficiency) of the proposed algorithm. The performance is demonstrated by synthetic and well-known benchmark functions.

**Audience:**

Yes

**Broader Impact Concerns:**

None.

**Claims And Evidence:**

Yes

**Requested Changes:**

I do not have critical concerns. Most of comments below are not significant issues.

- How the kernel hyper-parameters of all the methods were selected in the experiments? How can using the setting by Rando be justified?

- Figure 2 should be explained in more detail.
  - In the caption, Left and Right are lower and upper?
  - Why the bottom plot of Matern 3/2 is white?
  - What is the difference between quadratic (regular) and quadratic (canonical)?
  - square exponential and Matern is regular or canonical?
  - The definition of 'regular' should be explicitly clarified. (using Euclid distance?)

- In the proposed framework, is it valid to use the marginal likelihood of GP to select hyper-parameters? (though of-course it requires additional computational cost).

- Relation with the well-known DIRECT algorithm might be informative to discuss.

- For Sec6.1 (Experiments on synthetic functions), why time vs regret plot is not shown? Similar plots to Figure 7 should be informative for synthetic data as well.

- In Figure 6 and 7, why the initial values are different for different methods?

- Why the timing of finishing algorithms are different for each method? (both in iterations and wall-clock time)

- Although the authors repeatedly claim computational efficiency of the proposed method rather than sample efficiency, results on Figure 6 and 7 (evaluating sample efficiency and computational efficiency, respectively) are not largely different.

- Showing more detailed entire Algorithm with details such as the center selection, maximum distance calculation, and the partitioning procedure can be informative. Algorithm 1 is not specific for the proposed method. I first felt a bit difficulty to grasp the whole procedure of the proposed method.

- The original OO (without GP) should be a good baseline to demonstrate benefit of the proposed method.

Minor:

- In Abstract, readers cannot find what 'N' means in O(N log N).

- In introduction, after (Salgia et al., 2021), a missing period.

- In my opinion, the title for section 3 should be more about the proposed method (e.g, simply, 'Gaussian process optimistic optimization') rather than current 'Connection between Bayesian and Optimistic Optimization', because most of technical details of the proposed method are described entirely in section 3 (not only in section 3.4).

- What does 'budget N' means? The maximum number of iterations? (while in introduction, it is defined as the number of observed values).

**Strengths And Weaknesses:**

Strengths

- Computational efficiency of acquisition functions in BO should be a practically important topic.

- The proposed algorithm is simple to implement and seemingly easy to use.

Weaknesses

- Kernel hyper-parameter tuning are not provided, and sensitivity to kernel hyper-parameters is not clarified.

- Technical descriptions are somewhat cluttered.

---

### Review · Reviewer_qus2 · 2023-06-26

**Summary Of Contributions:**

This work presented an optimization method under the optimistic optimization (OO) framework where the key idea is to partition the search space by a binary tree. The connection to GP comes in with the usage of canonical pseudo-metric (depends on the kernel function), which is used to bound the probabilities of $P(|f(x)-f(y)|>u)$ (Section 3.1, Pisier (1999), Theorem 4.7) and further an upper bound of a cell (Section 3.2). The upper bound of a cell is used in the OO framework to select which cell to expand.

The authors provided theoretical analysis of the upper bound of the cumulative regret (Proposition 1) and convergence rate of regret (Proposition 2). Besides, experiments on synthetic functions shows the proposed method is competitive, especially in the computation time.

**Audience:**

Yes

**Broader Impact Concerns:**

No concerns.

**Claims And Evidence:**

No

**Requested Changes:**

I appreciate the connections that the authors draw between GP prior and OO, and also the theoretical analysis. But I  think without including the estimates of the GP hyerparameters, the proposed method can be hard to apply in practice.

The experiments are all on the synthetic functions, which make it less convincing. Also the experiment detail is missing and I find it hard to follow what is going on.

A whole series work of scalable Gaussian Process is not discussed in the relevant work, nor in the evaluations.

**Strengths And Weaknesses:**

### Pros

The strength of the work lies in 1) discovering that by using canonical pseudo-metric, one can estimate the upper bound of a cell with a GP prior, thus connecting OO and BO. 2) The author also provided theoretical analysis on the upper bound and convergence rate cumulative regret. 3) The paper is mostly clearly written (except experiments).

### Cons

The biggest weakness of the work is to require knowing the ground truth GP hyerparameters, which significantly limits the impact of the work. This is also mentioned by the authors in their Conclusion “On the other hand, the strong reliance on the prior assumptions without being able to adapt the hyperparameters of the prior on the fly, is currently still a limitation in practical settings”. Without this problem solved, I can’t see how the proposed method can be useful in practice.

Second, I am not sure I understand how the authors created the plots. For example, in Figure 6 and 7, what is the budget applied to all the methods? For me it should be either all the methods are allocated with the same time or with the same number of evaluations. But in both Figure 6 and 7, the methods all end up at a different budget. Why is that? Also, the experiments on benchmarks are also synthetic functions, which is a bit weak by current evaluation standards. More importantly, without real-world examples, it is hard to advertise why the setting that the authors study, is an important one. In the end, more detail or explanation about experiment setting will be helpful.

Besides TurBO, there are some relevant works on scalable gaussian processes:
- Wang, Ke, et al. "Exact Gaussian processes on a million data points." Advances in neural information processing systems 32 (2019).
- A. G. d. G. Matthews. Scalable Gaussian process inference using variational methods. PhD thesis, University of Cambridge, 2016
- M. K. Titsias. Variational learning of inducing variables in sparse gaussian processes. In AISTATS, pages 567–574, 2009.
- J. Hensman, N. Fusi, and N. D. Lawrence. Gaussian processes for big data. In UAI, 2013


Minor questions:
- Why in Figure 4 that TurBO had a sudden drop? And why is there a spike for GP-OO around 700 iterations?
- Figure 2: Right columns, Left columns -> Bottom, Top columns? One plot of cells is all white color?
- The left plot in Figure 4 and right plot in Figure 5 are of black and grey colors, for me it is harder to read the plot.

---

### Decision · Action_Editors · 2023-09-09

**Recommendation:** Accept as is

**Comment:**

All reviewers were satisfied with the clarifications provided by the authors and vote acceptance.

**Audience:**

The paper is suitable for the TMLR audience.

**Claims And Evidence:**

Claims are supported. The proposed method is sound and has a theoretical backing. All reviewers were satisfied with the clarifications and found that this work makes good contributions.